# The Dynamic Evolution of Global Energy Security and Geopolitical Games: 1995~2019

**DOI:** 10.3390/ijerph192114584

**Published:** 2022-11-07

**Authors:** Getao Hu, Jun Yang, Jun Li

**Affiliations:** School of Economics and Business Administration, Chongqing University, Chongqing 400030, China

**Keywords:** energy security, dynamic evolution, geopolitical game

## Abstract

Under the influence of economic globalization, the internationalization trend of energy security issues has become increasingly prominent. This paper adopts the natural discontinuity grading method to classify the energy security status of 102 countries into five categories: Best, Better, Good, Poor and Worse types; reveals the dynamic evolution characteristics and main formation mechanisms of world energy security; and puts forward the game focus of future energy geopolitics. The results show that: (1) during 1995–2019, global energy security presents local turbulence and an overall “J” shaped trend; (2) the global energy security pattern coincides with the international geopolitical order. The countries with the “Best” energy security are found in Western Europe and North America while the countries with “Poor” or “Worse” energy security are located in Asia and the less developed regions of Africa; (3) the main reason why developed economies have better energy security is due to their high energy use efficiency, while developing countries lag behind mainly because of their lower innovation capacity, lower productivity and lower disposable income; and (4) the global energy security landscape is expected to be affected by the changing US–China relationship, coercive energy transition and the uncertainty of the political environment.

## 1. Introduction

As an essential factor in economic and social development, energy has profoundly affected geopolitical stability [1]. Since the 1970s, abundant hydrocarbon resources have led to many energy-related wars, political and economic conflicts and global financial crises, which have exacerbated the instability of energy prices [2,3]. In recent years, energy security has become a serious public issue, thus receiving the most international attention in research and policy [4]. In addition to rapidly rising global populism, unilateralism and protectionism are gradually breaking the rules of international market competition and reshaping geopolitical patterns, further challenging economic globalization, changing the global energy security pattern and affecting the governance system significantly. Moreover, geopolitics will play a crucial role in determining whether countries with relatively low energy self-sufficiency have access to energy commodities [5]. Undoubtedly, energy security, as a political weapon, will contribute more to economic growth and social welfare [5,6]. Given the volatile nature of the international political scene and the close link between supply and demand in the energy market, it is essential to study global energy security and how it is changing. This will allow us to better understand energy diplomacy and global governance.

In geopolitics and energy geography, mitigating resource or energy security concerns have become a top priority. The cognitive boundaries and connotations of the conception of “energy security” have been broadened and deepened by researchers and policymakers, yet remain inconsistent. In the early 1970s, the International Energy Agency (IEA) posed the concept of energy security as “uninterrupted energy supply at affordable prices” in response to the impact of the oil crises on the security of crude oil supply. In subsequent studies, the security of energy supply has been extensively used as the most critical aspect of energy security. For instance, as defined by Bohi et al. [7] and Bielecki [8], “energy security” refers to the security level of capacity of a national or regional energy system to adequately supply energy at reasonable and acceptable prices. According to the Asia Pacific Energy Research Center (APERC), energy security refers to the ability of the economy to ensure that energy can be obtained sustainably and promptly and that the level of energy prices does not interfere with the average economic performance of the economy. Based on the aforementioned concepts, APERC put forward the 4A conception of energy security, namely availability, acceptability, accessibility and affordability [9]. With the increasingly catastrophic consequences of extreme climate events and the growing concerns about global warming and air pollution [10,11,12,13], it has been widely acknowledged that the nexus between environmental sustainability and energy consumption is important. For example, considering the importance of sustainable energy use, many scholars have incorporated it into the framework of energy security assessment [14,15,16,17]. In addition, great progress has been made in making the concept of “energy security” more people-oriented and social and the levels of energy services and economy and technology have become the new dimensions of energy security. Lesbirel [18] and Vivoda [19] pointed out the crucial role of the national energy service level in energy security. However, it is worth noting that energy security reflects the balance of local energy supply and demand and involves underlying games of political relations evolution and development rights. It follows that the conceptual scope of “energy security” has gradually extended to the field of geopolitics, forming a scientific proposition covering global economic and social development and political and diplomatic relationship. For instance, based on the study of Vivoda [19], Sovacool [20] extended the conception of energy security from 20 dimensions. Wang et al. [21,22] also estimated the regional energy security level in terms of three aspects: energy supply-transmission security, energy use security and social stability. Diversification plays a key role in determining energy availability and security. Meanwhile, the infrastructure related to energy conversion and transmission is essential to stabilize the energy supply and promote economic security [23,24,25,26].

In the past, scholars often put forward energy security assessment frameworks when studying the issues related to energy security. For instance, based on 25 indicators from three dimensions, Martchamadol et al. [27] proposed a comprehensive energy security performance indicator (AESPI) to estimate the energy security level in Thailand. Based on ten indicators from four dimensions, Sovacool et al. [28] built an energy security evaluation framework to assess the long-term changes of energy security performance (ESP) of the United States and 22 European Union (EU) countries. Furthermore, Sovacool et al. [29] enlarged the dimensions and indicators to 5 and 20, respectively and conducted an empirical study of 18 countries. Stavisky et al. [3] constructed an assessment framework of energy security consisting of six dimensions and 28 indicators and evaluated the energy security status for 49 European countries. Using the functional data analysis method, Bamisile et al. [30] analyzed the dynamic evolution of the Chinese energy security index, which consists of four dimensions and 12 indicators. These studies show that apparent disparities in establishing energy security dimensions exist among studies and the selection of indicators corresponding to each dimension also differs substantially. Apparently, the availability of data and the differentiated conception and dimension associated with energy security can significantly impact the selection of energy security indicators. Indicators commonly used in existing literature include energy consumption per capita, energy intensity, energy self-sufficiency, electricity prices, electricity supply, carbon intensity, CO_2_ emissions per capita and CO_2_ emissions [31]. In addition, some studies have introduced specific indicators to take into account the special characteristics of the research objects. In different studies, indicators are usually assigned to different dimensions. For example, “energy intensity”, as one of the most commonly used indicators, has been assigned to different dimensions such as economy [25], technology and efficiency [32,33], energy demand [34,35] and security of supply [36].

Energy security research can broadly be divided into two areas: studies that look at the issue from a spatial perspective and studies that examine it from a temporal perspective. In terms of spatial studies, most of them have merely focused on a single country [37,38,39], countries in specific regions [28], economic organizations [40,41], neighboring countries [25,42] and island countries [43,44], failing to provide abundant evidence from a global perspective. In terms of temporal studies, the time span in previous studies was usually divided into single years [27,45], discrete years [22,46] and consecutive years [47] and few have provided long-term continuous annual analysis. As far as we know, few studies have assessed the energy security of countries worldwide (i.e., more than 100 countries) for more than 20 consecutive years. Regarding the methodologies, principal component analysis (PCA) has been widely utilized to analyze the relationship between variables [48]. PCA can effectively extract key information from the original data by using a multivariate technique. Additionally, data envelopment analysis (DEA) has been the mainstream performance assessment method [49]. Similarly, assurance region (AR) methods have also been widely used to solve the problem of the arbitrary setting of inputs/outputs weights in the DEA model [18]. Wu et al. [50] have proposed the integrated PCA/DEA-AR method to estimate the energy security of each country investigated, avoiding the generation of optimal weights equal to zero. Regarding the weights of the associated indicators, many scholars have used weighting procedures, such as weighting based on expert opinion. However, this method has subjective solid arbitrariness. According to Gasser [51], equal weight assignment is the most popular method, which may be attributed to its simplicity. However, it is sometimes difficult to capture the critical differences between indicators, or it may mask the lack of a theoretical basis for weight determination. In recent years, a growing number of researchers have been establishing more objective weighting methods based on mathematical knowledge, including the eigenvector method, the weighted least square method and the entropy weighting method. These methods determine the weights automatically through mathematical models without considering the subjective preferences of decision makers [30].

Based on previous literature, some conclusions can be drawn. (1) Extensive use of energy supply, use, economic and environmental dimensions have allowed for a comprehensive evaluation of energy security performance among countries; (2) spatially, the focus of these studies is on how energy security affects individual countries, economic organizations and specific regions.; temporally, the sample period is discontinuous and rarely spans more than 20 years [52]. (3) Subjective weight assignment is the most popular method [20,53], while more objective weight assignment methods need to be continuously explored. There are still substantial future opportunities to develop assessment methods and select relevant dimensions and indicators. When it comes to establishing assessment models for energy security, researchers have mostly relied on principal component analysis [39,54,55]. This means that new research methods for assessing energy security need to be constantly explored.

This study contributes to relevant literature along three dimensions. First, the previous studies associated with global energy security assessment mainly focused on the situation in a single period, failing to reflect the changing trend and global challenges of energy security. Hence, we constructed a comprehensive regional energy security index and used the entropy method to measure the level of energy security for 102 countries from 1995–2019. Second, a limited number of past studies have looked at energy security from a worldwide perspective. This study used functional data analysis to comprehensively and intuitively analyze the potential changes in energy security from a dynamic perspective. Through the functional processing of discrete energy security data, we analyzed the difference in energy security level among different regions. Third, affected by resource endowments, there are apparent disparities in the energy security status of different countries. In order to circumvent the potential bias in results associated with using non-parametric estimation methods, this study employed the kernel density estimation method to analyze the spatial variation and dynamic evolution of the distribution. It captures the distribution of each dimensional index, thus suggesting the variability of each energy security dimension. Therefore, it is well placed to explore the mechanisms driving the evolution of energy security patterns across 102 countries worldwide.

This study aims to estimate the energy security status of 102 countries comprehensively and systematically. First, we adopt a functionalized data analysis method to form dynamic continuous curves from discrete data. Compared with discrete data analysis methods, the FDA focuses on the analysis of curved data without assuming any particular distribution of the data, which is rarely used in existing studies. Then, by using the Jenks natural discontinuity grading method, this study classified the 102 countries into five categories to reduce the subjectivity effect in classification. Finally, this study uses the method of kernel density estimation to assess the variability of each energy security dimension, which has the advantage of not assuming that the data follows any particular distribution. This research framework yields interesting results and deepens the understanding of global energy security.

The rest of the paper is organized as follows. Section 2 details the data and methodology. Section 3 presents the dynamic evolutionary process of energy security. Section 4 analyzes the disparities of global and regional energy security and Section 5 summarizes the conclusions and discusses policy implications.

## 2. Methodology and Data

### 2.1. Entropy Weight Method

In previous studies, the commonly used evaluation methods such as the Delphi method [56] and the fuzzy comprehensive evaluation method [57] often have relatively high subjective ingredients. Compared with those methods, the entropy weighting method based on the information entropy of indicators can fully use the specific relationship between the difference between the indexes and that of the information contained in the original data. By adjusting and correcting the objective weights of the indexes according to the variability of the indexes, it further ensures the maximum utilization of the original data and minimizes the influence of subjective factors as much as possible. The dispersion degree of index data represents the difference between individuals or the difference in the effective value of information, further quantitatively determining the final weight of evaluation indicators. On the whole, the entropy weighting method can maximize the utilization of the original data information. Therefore, the entropy weight method is used to calculate the energy security index of 102 countries, as follows.

Step 1 The initial data matrix *X* consisting of m evaluation objects and *n* evaluation indexes is standardized using the polarization transformation method.

For positive indicators in which a larger value is better, such as the industrialization level.
(1)$$x_{ij}^{’}=\frac{x_{ij}-min\left(x_{ij}\right)}{\max\left(x_{ij}\right)-min\left(x_{ij}\right)}$$

For negative indicators in which a smaller value is better, such as electricity per capita.
(2)$$x_{ij}^{’}=\frac{\max\left(x_{ij}\right)-x_{ij}}{\max\left(x_{ij}\right)-min\left(x_{ij}\right)}$$

Step 2 Calculate the proportion $y_{ij}$.
(3)$$y_{ij}=\frac{x_{ij}^{’}}{\sum_{n}^{m}x_{ij}^{’}},j=1,2,\ldots ,n$$

Step 3 Calculate the information entropy $e_{j}$.
(4)$$e_{j}=-k\sum_{i=1}^{m}y_{ij}lny_{ij},j=1,2,\ldots ,n$$
where $k=\frac{1}{lnm}$ is a non-negative constant and $0\leq e_{j}\leq 1$; specifically, when $y_{ij}=0,y_{ij}lny_{ij}=0$.

Step 4 Calculate the difference coefficient $d_{j}$.
(5)$$d_{j}=1-e_{j},j=1,2,\ldots ,n$$

Step 5 Calculate the weight.
(6)$$\omega_{j}=\frac{d_{j}}{\sum_{j=1}^{n}d_{j}}=\frac{1-e_{j}}{n-\sum_{j=1}^{n}e_{j}},j=1,2,\ldots ,n$$

Step 6 Calculate the energy security index (*ESI)* value in scenario.
(7)$$ESI_{i}=\sum_{j=1}^{n}y_{ij}\omega_{j},i=1,2,\ldots ,m$$

### 2.2. Smoothing of Discrete Data

As a statistical technique, functional data analysis (FDA) has been widely applied to the collection of functional data. In contrast to traditional statistics, which usually use numbers or vectors, the FDA utilizes curves or graphs as data units. This means that the FDA can be used to conduct analysis based on the curvilinear form. Many multivariate statistical methods can be applied directly to the functional data, but the FDA can explain more information about the observed function, formulated as follows.
(8)$$x_{i}\left(t\right)=\sum_{k=1}^{K_{i}}c_{ik}\varphi_{k}\left(t\right),i=1,2,\ldots ,m$$
where $\left\{\varphi_{1}\left(\cdot \right),\ldots ,\varphi_{K_{i}}\left(\cdot \right)\right\}$ denotes the set of basic functions and $\left\{c_{1k},\ldots ,\varphi_{ik}\left(\cdot \right)\right\}$ stands for the set of corresponding coefficients. In this paper, we assume that $\varphi_{k}$ is a B-sample basis function and $c_{ik}$ is the corresponding coefficient. Meanwhile, this paper allows for using the same basis function in all samples while setting a different number of bases. Then, the nearest similar coefficients can be estimated by spreading the basic functions. The least squares method is usually used for estimation.
(9)$$min\left\{SSE=\sum_{j=1}^{J}{\left[x_{i}\left(t_{j}\right)-\sum_{k=1}^{K_{i}}c_{ik}\varphi_{k}\left(t\right)\right]}^{2}\right\}$$

Then, the sum of squared residuals is penalized using the minimization method with the following equation.
(10)$$PENSSE\left(\lambda \right)=\sum_{j=1}^{J}{\left[x_{i}\left(t_{j}\right)-\sum_{k=1}^{K_{i}}c_{ik}\varphi_{k}\left(t\right)\right]}^{2}+\lambda \int {\left[Dx_{i}^{m}\left(t\right)\right]}^{2}dt$$
where *$\lambda \int {\left[Dx_{i}^{m}\left(t\right)\right]}^{2}dt$* indicates the rough penalty term used to measure the degree of smoothness of the function $x_{i}\left(t\right)$; *m* refers to the order of the derivative, which is usually taken as two to satisfy the general problem; *λ* is the smoothing parameter. In the framework of the basis function, *λ* is a parameter vector whose value can be verified by cross-validation (CV) or generalized cross-validation (GCV).

### 2.3. Nature Discontinuity Point Grading Method

Based on the energy security index of 102 countries derived from the entropy weighting method, this paper uses the natural discontinuity point grading method to classify the sample countries into five categories: Best, Better, Good, Poor and Worse types. On the basis of the inherent natural grouping in the data, the natural discontinuity grading method can identify the classification intervals, further appropriately grouping similar values and maximizing the differences among classes. This grouping method divides the data into classes by setting class boundaries where the differences in data values are relatively large.

Step 1 Calculate the sum of squares of total deviations (*SDAM*) of each class.
(11)$$SDAM=\sum {\left(X_{i}-\bar{X}\right)}^{2}$$
where $\bar{X}=\frac{1}{n}\sum_{i=1}^{n}X_{i}$, *n* denotes the number of elements in the array and $X_{i}$ represents the value of the element.

Step 2 The sum of squares of total class deviations (SDCM) is calculated for each combination of ranges in the classification result and the minimum value is denoted as *SDCM_min_*. The elements are divided into *k* classes, so that the classification results can be divided into *k* subsets, one of which is $\left(X_{1},X_{2},\ldots ,X_{i}\right)$, *$\left(X_{i+1},X_{i+2},\ldots ,X_{j}\right)$*, …, $\left(X_{j+1},X_{j+2},\ldots ,X_{n}\right)$. Then, the sum of squares of total deviations of the first class, *SDAM*_1_ can be calculated as follows:(12)$$SDAM_{1}=SDAM_{i}+SDAM_{j}+\ldots +SDAM_{n}$$

Similarly, the classification results can also be divided into other cases of *k* classes. The values of $SDCM_{2}+\ldots +SDAM_{c_{n}^{k}}$ are calculated sequentially and the minimum value is chosen as the final end *SDCM_min_*, so that this classification range is the best classification.

Step 3 Perform validation by the goodness of fit and calculate the gradient $gvf_{i}$ of the class as follows:(13)$$gvf_{i}=\frac{SDAM-SDCM_{i}}{SDAM}$$

$gvf_{i}$ ranges from 1 (perfect fit) to 0 (poor fit), with a higher gradient indicating greater difference among classes. The gradient value of the classification obtained by the *SDCM_min_* in step 2 is the largest, implying that the results derived from the natural discontinuity grading method is desirable.

### 2.4. Kernel Density Estimation Method

The kernel density estimation method, being a nonparametric testing method, has had extensive use in studying the spatial variation and dynamic evolution of distributions. This method also has the advantage of preventing subjectivity in nonparametric estimations when choosing functions and guarantees the continuity of the estimated density curves. First, we apply the above method to estimate the density of the global energy security index and its three main energy sub-dimensions for 102 countries. Then, we plot the density curves to identify the main driving mechanisms of the evolution of global energy security patterns over the past 25 years in a more intuitive and concrete way. The specific model is as follows.
(14)$$f\left(x\right)=\frac{1}{nh}\sum_{i=1}^{n}K\left(\frac{x-x_{i}}{h}\right)$$
where *n* represents the number of samples, *h* denotes the window width and K(·) shows the kernel function. The frequently used kernel functions include quadratic kernel, Gaussian kernel and uniform kernel, of which the first two are the most popular. Considering the relatively small impact of selecting the kernel function, this paper adopts Gaussian kernel.
(15)$$K\left(x\right)=\frac{1}{\sqrt{2\pi }}exp\left(-\frac{x^{2}}{2}\right)$$

The window width is the parameter that will determine how accurate the kernel density estimation is and how smooth the density map looks. Therefore, determining the appropriate window width has a direct impact on the results of kernel density estimation.

### 2.5. Data Collection

In this paper, we used a global dataset for the period from 1995 to 2019 and constructed the energy security evaluation index system from four dimensions, namely energy supply [34,36], energy use [34], energy economic development [32,34] and energy environment [34,36] (Table 1). The energy supply dimension describes and evaluates the capacity of the regional energy system to cope with the imbalance of the energy supply–demand relationship, which contains six indicators: energy self-sufficiency rate [34,36,58], power generation to energy consumption ratio [21], renewable power generation to total power generation ratio [3,21], non-fossil fuel consumption ratio [21,58,59], fuel imports to energy consumption ratio [3] and energy production per capita [34,58]. The energy use dimension measures regional energy consumption and efficiency. Energy use security assesses the energy security level from the energy demand perspective. Greater energy demand associated with higher energy consumption puts a tremendous burden on energy supply. So eight indicators are chosen for this dimension: electricity intensity [60], energy consumption per capita [21,58], energy intensity [21,32], electricity consumption per capita [30], energy consumption elasticity coefficient [36,61], energy population elasticity coefficient [34,61], energy losses [3] and energization rate [50]. The energy economic development dimension evaluates the economic development level of a country. The higher the level of economic development, the stronger the ability to bear energy costs. It can partially offset the adverse effect of price factors on energy security. This dimension includes six indicators: level of industrialization [59,61], level of urbanization [31,61], percentage of population growth [61], GDP per capita [21,30,58], coefficient of elasticity of electricity production [61] and GDP growth rate [34]. The energy environment dimension evaluates the ability of regional energy use to impact the environment. So far, coal and oil have dominated the global energy consumption mix. However, the use of fossil resources has had a negative impact on the environment. Environmental security has received wide attention from scholars. This dimension contains four indicators: nitrogen oxide emissions per capita [20], carbon intensity [30,58], share of particulate emissions [3] and share of clean energy [58,59].

All indicators selected for this study are shown in Table 1. CO_2_ intensity, energy depletion, particulate damage emissions, percentage of population growth, GDP per capita, industrialization level, urbanization level, electricity energization rate and NOx emissions per capita were obtained directly from the World Bank. Other energy-related indicators’ data was retrieved from the energy information administration of the United States. Fossil energy consumption mainly includes coal, oil and natural gas.

## 3. Main Results

### 3.1. Assessment of Global Energy Security

In this section, the energy security index of 102 countries from 1995 to 2019 is measured quantitatively using the entropy weight method. Then, by using the natural discontinuity point grading method, the energy security status can further divide into five categories: Best, Better, Good, Poor and Worse. The distribution of the energy security status is shown in Figure 1.

During the sample period, the global energy security landscape has been gradually optimized (Figure 2). The number of “Best” energy security countries shows a significant upward trend, whereas that of countries with “Worse” energy security decreases significantly. From 1995 to 2018, countries with “Best” energy security increased from 4 to 11, while that of countries with “Better” performance increased from 5 to 18. The number of countries with “Good” energy security has remained stable. Conversely, the number of “Worse” and “Poor” types showed a downward trend, falling from 26 and 46 to 14 and 28, respectively. In 2019, threatened by the outbreak of neo-coronavirus pneumonia, the number of countries with energy security “Best”, “Better” and “Good” types decreased by 2, 3 and 7, respectively, while that of countries with “Worse” and “Poor” energy security increased by 1 and 12, respectively.

However, considering the different stages of development, the global energy security landscape enjoyed overall stability in the last five years of the 20th century, while that in 2010 underwent significant changes. From 1995 to 2010, the number of “Best” and “Better” energy security countries increased significantly, with the two types increasing from 9 to 17. Meanwhile, the number of countries with “Worse” and “Poor” energy security decreased slightly. In conclusion, before 2010, the overall optimization of the global energy security pattern was not obvious, even showing a trend of local deterioration (Table 2). However, from 2010 to 2018, along with the implementation of global sustainable development planning and sustainable governance systems, as well as the long-term low global crude oil prices, the number of “Best” and “Better” type countries increased, whereas that of “Good”, “Poor” and “Worse” countries declined significantly. In 2019, the global energy security situation worsened due to the impact of global neo-coronavirus pneumonia.

### 3.2. Evolution of Global Energy Security

The discrete energy security scores are functionalized to analyze the potential dynamic evolution of energy security. As shown in Figure 3, the level of energy security varies across countries during the sample period, varying from 0.1 and 0.3 in most countries. Obviously, the energy security levels of countries have changed a lot. In recent years, the overall level of energy security has improved and Norway ranks first globally. Figure 4 shows the mean value function curve of global energy security, which roughly reflects the general trend of global energy security. As shown in Figure 4, global energy security is divided into seven stages from 1995 to 2019, which is consistent with the global macroeconomic development and energy market changes. From 1995 to 1996, global energy security had improved after the end of the Gulf War. From 1997 to 1998, the Asian financial crisis led to an overall decline in global energy security. From 1999 to 2000, energy security continued to improve. From 2001 to 2004, global crude oil prices rose and global energy security declined again. From 2005 to 2006, energy security continued to improve. From 2007 to 2009, the global financial crisis broke out, leading to a decline in global energy security. After 2010, countries became more environmentally conscious and energy security continued to improve. On the whole, global energy security presents local turbulence and an overall “J” shaped trend.

### 3.3. Robustness Tests and Heterogeneity Analysis

The key to constructing a composite index lies in the weights and aggregations of the indicators to form the final results. A clear understanding of DESI(t)’s strengths can be had by contrasting it with other commonly used weighting and aggregation methods. This section uses an approach of deleting indicators, focusing on how the uncertainty in quantifying indicator scores can be filtered through the structure of the composite index to filter out this structure, thus ensuring that the synthetic assessment results are sound. As a result, this section no longer contains population growth rate, share of renewable energy generation and GDP growth rate. The remaining 21 indicators were re-weighted and aggregated (Figure 5). As shown in Figure 5, both curves can effectively capture the worst-case scenario of global energy security around 1995 to 2010 and predict a decline by 2019 due to the impact of the global epidemic. Meanwhile, both curves showed a significant upward trend, indicating that the evaluation results and fluctuation of the two curves are essentially consistent.

Based on the data of net energy import, this study divides global countries into energy importers and energy exporters. The former includes 68 importing countries with positive net energy imports, while the latter includes 34 exporting countries with negative net energy imports. As shown in Figure 6, during the sample period, the energy security of importing countries has shown an increasing trend, whereas that of exporting countries has shown a decreasing trend. However, compared with importing countries, exporting countries consistently have higher levels of energy security. Countries with large fossil fuel resources tend to export energy, while their economies are largely based on mineral resources and heavy industry. They often have high pollution emissions, low industrial energy use efficiency and high greenhouse gas emissions. These countries are also prone to geopolitical conflicts and economic turmoil. In fact, since the 1990s, the global energy crisis has been associated with these hydrocarbon-rich countries. It makes sense, therefore, that the energy security index for exporting countries is gradually declining. Conversely, energy importers are concentrated in countries with advanced economies. These country’s strong political and economic foundation creates an ideal environment for developing energy systems and increasing energy security. Meanwhile, these countries continue to strengthen their leadership and organizational position in the global governance system and develop clean energy through a range of programs. Therefore, the energy security index of energy importing countries shows an upward trend.

### 3.4. Evolution of Regional Energy Security

This section dynamically evaluates the energy security levels of 102 countries from 1995 to 2019 and analyzes its evolution process, showing that the global energy security pattern largely coincides with the international geopolitical order.

(1) The “Best” countries are concentrated in Western Europe and the economically developed regions of North America. Despite the fact that energy resources are not very plentiful, economic globalization has spurred developed countries in Western Europe to obtain energy from Russia and the Middle East through the use of strategic energy reserves and energy diplomacy, so as to maintain a stable energy supply. Additionally, Western European countries improved their levels of national energy use security by strengthening their leadership and organizational position in the global governance system and adopting a series of “climate and energy plans” to vigorously promote clean energy development and greenhouse gas emission reduction campaigns. Specifically, the number of “Best” countries in Western Europe increased from one in 1995 to four in 2019, making it a concentrated area of “Best” energy security countries (Figure 7b). Unlike the performance of the energy systems of the European economies, the energy-rich United States and Canada have a high level of energy supply security, whereas the environmental sustainability of energy use is relatively low due to the rapid development of their fossil industries in recent years (Figure 7a). Since 1995, along with their low fossil energy reserves, both countries have shown a relatively small decrease in the energy supply security index, while their energy use security index has shown an increase due to the energy transition and development of a low carbon economy. Overall, their security level is still significantly lower than the EU economies. However, the stable political environment and strong economic support have created even more favorable conditions for them to improve energy security level and their energy systems.

(2) The countries with “Better” energy security are mainly located in central Europe, Latin America, central Africa and Asian high-income countries. The formation and development of global energy markets have, to some extent, reduced the dependence of regional energy security on local resources. The economic globalization since 1995 has resulted in a shift of “Better” countries with advanced productivity to Central and Eastern European countries in the “second tier” of the European Union, high-income countries in East Asia and energy-rich countries with energy resources. In 1995, the five safer countries were concentrated in the fossil energy-rich Middle East and Southeast Asia and in Latin America, where the level of hydropower development is high (Figure 8a). In these countries, the ratio of the index of Energy Supply Security to that of Energy Use Security has fluctuated wildly, ranging from 2.5 to 11.5 and reflecting a significant gap between energy supply and demand. However, with the progress of global energy development technology and a deeper understanding of climate change and low-carbon economy, global countries have performed better in the energy security use index. The data from 2019 shows that the 15 energy-secure countries had a significantly lower ratio of energy supply to demand, from 1.2 to 6.5 (Figure 8b). This indicates that the issue had been resolved to a certain extent.

After the dramatic changes in Eastern Europe, Bulgaria, Poland, Lithuania and other Eastern European countries joined the EU and undertook the outward transfer of the manufacturing industries of Western Europe, relying on their relatively cheap labor and relaxed production environment. With rapidly changing economic patterns, these countries have gradually improved their energy use efficiency and environmental sustainability and have witnessed the continued growth of the index of each energy security dimension during the sample period (Figure 9a). However, limited by the domestic economic level, innovation capacity and institutional mechanisms, the energy security supply index and economic development index are significantly lower than those of the developed EU economies. In 2019, Africa had four countries with “Better” energy security. Despite their backward economy, some African countries have accelerated their industrial restructuring in recent years and made full use of abundant natural resources and per capita energy consumption far below the level of developed countries, making their environment security index higher than other countries. In addition to this, the energy security usage index and economic technology index are both increasing as energy demand rises. Although the overall energy security index fluctuates, the overall total index value is consistently higher (Figure 9b). For example, Angola exports 50% of its annual oil production to China, making it one of the top ten countries that export oil into China.

The Latin American region has abundant hydropower resources, with hydroelectric power generation accounting for approximately 65% of total regional electricity generation, resulting in a high energy security supply index and usage index (Figure 10a). Between 1995 and 2010, the energy security index decreased steadily as a result of political instability, economic recession and deteriorating energy development conditions. In addition, the energy security economic development index is relatively low due to the higher cost of hydropower and higher average prices of regional electricity. After 2010, the overall energy security index for the Latin American region increased rapidly. Conversely, relying on abundant hydrocarbon resources, countries such as the UAE and Saudi Arabia have higher regional energy availability and lower energy prices, thus resulting in a higher energy security economic development index (Figure 10b). However, these countries depend primarily on mineral resource exploitation and heavy industry and have higher energy consumption and pollution emissions, so their energy security use index is lower than their supply index and economic development index. Meanwhile, caused by the decreased energy reserves of each country and the homogenization of energy supply types, these countries’ energy supply security index has decreased from 0.22 to 0.12 since 1995. The world’s largest coal exporter, Australia remains a net importer of crude oil and its energy security supply index shows a significant downward trend due to its decreasing energy reserves (Figure 10c). Meanwhile, the energy security index of Australia has increased significantly since 2005, as the government has made climate change the most important political issue and actively transformed its economy. In addition, compared with Latin America and hydrocarbon-rich countries, the energy environment index of Australia has shown an increasing trend.

(3) The countries with “Good” energy security are mainly located in the Middle East, Southeast Europe and East Asia. Restricted by the political environment, economic level and modernization process, the security of regional energy system in these countries is facing challenges or undergoing rapid transformation. The Middle East, for example, has rich hydrocarbon resources, high levels of energy supply and advanced economic technology, but their economy is largely driven by the petrochemical industry, resulting in low industrial energy efficiency and high greenhouse gas emissions. Meanwhile, coupled with the continuous political conflicts and economic turmoil, their energy security use index, environmental index and economic development index all lag behind the supply index and the overall energy security index shows a downward trend (Figure 11a).

After the dramatic changes in Eastern Europe, Ukraine, Albania, Turkey and Russia joined the Black Sea Economic Cooperation (BSE), to some extent, stabilizing energy markets, diversifying energy structure and stabilizing domestic energy prices. Generally speaking, the energy security index fluctuates in some years, whereas the energy security index in all dimensions and the overall energy security indexes shows an increasing trend. (Figure 11b). In addition, considering the high concentration of fossil energy in Russia and the strong dependence of other member states on Russia’s energy supply, the potential threat to the energy supply security of this region is relatively high. Meanwhile, these countries still retain the economic pattern of the former Soviet Union, which leads to an irrational industrial structure, extensive energy use, low penetration of modern renewable energy technologies and hence a lower overall energy security index than the EU countries.

Since the 21st century, the rapid rise of emerging East Asian economies, such as China, Indonesia, Thailand and Malaysia, has led to continuous growth in the energy demand index and decline in the energy security supply index and environment index. Since 1995, with rapid economic development, developing countries have actively responded to the UN Sustainable Development Goals, strengthened energy infrastructure and basic services capacity building and vigorously eliminated energy poverty. Therefore, their energy economy development index has significantly increased (Figure 11c). For example, Indonesia’s energy economy development index doubled during the sample period. Meanwhile, rapidly increasing energy consumption enhanced its dependence on the foreign market, making the regional energy security supply index appear to show an upward trend. For example, as the world’s largest energy consumer and greenhouse gas emitter, China’s dependence degree on oil and natural gas imports reached 70.8% and 43% in 2019.

(4) The countries with “Poor” and “Worse” energy security are concentrated in the Asian and African regions. In the global economic landscape, due to lagging social productivity, a hostile development environment for energy systems, poor infrastructure and public services, African and South Asian countries have a low level of energy economic development. In addition, coupled with the single energy supply structure and unsustainable energy use, these countries performed poorly in overall efficiency (Figure 12b). Given the increasing energy demand and insufficient supply, the overall energy security index did not grow until 2015. However, as the least energy secure region in the world, Africa contained five at-risk countries in 2019, accounting for one-third of the total number of “Worse” energy security countries worldwide. With frequent ethnic and religious conflicts, relatively poor energy resource endowments and backward production capabilities, the energy security of countries, including South Asia’s India, Nepal, Pakistan, Sri Lanka, Bhutan, the Philippines, Myanmar and Cambodia, has been in a dangerous situation for a long time. On the whole, the energy security composite index fluctuated considerably during the sample period and current energy security is likely to deteriorate further (Figure 12a). Since 2010, along with the popularization of electric power facilities, the implementation of oil and gas energy subsidy policies and the proliferation of distributed renewable energy technologies, the energy economy and technology level in South Asia has improved significantly. For example, Cambodia had an access rate of only 16.22% in 1995, while the access rate exceeded 84% in 2019, for which the energy economic technology index increased by more than two times.

As high-income Asian countries, Japan, South Korea and Singapore perform less well on the energy security supply index, resulting from their high foreign energy dependence. For example, the Fukushima nuclear crisis has gradually increased the proportion of Japan’s net energy import in total energy consumption from 80% to 94%, indicating that the security of energy supply has been seriously challenged. Meanwhile, the energy security supply index decreased from 0.026 in 1995 to 0.023 in 2019. Although Singapore gets more than 90% of its electricity from imported natural gas, the rapid development of solar energy and the liberalization of the electricity market have significantly improved its energy supply security. The associated index subsequently increased from 0.003 in 1995 to 0.005 in 2019, yet the overall energy supply capacity remained low (Figure 12c).

## 4. Exploration of the Driving Mechanism

### 4.1. Two-Dimensional Kernel Density Estimates for Global Energy Security

Based on the above analysis, this section uses the Kernel density estimation to evaluate the energy security level of 102 countries from 1995 to 2019 and further plots the Kernel two-dimensional graphs (Figure 13 and Figure 14).

As shown in Figure 13, the global energy security level differs substantially and does not obey the regular empirical distribution. On the whole, the energy use index, energy economy index and energy environment index are characterized by unimodal or multimodal distribution, suggesting obvious gradations of the three major dimensions of the global energy security level. Among them, the energy use index is characterized by unimodal distribution and no obvious grading trend appears. However, its peak value is still low, indicating a low level of development of the energy security use index, with a few years in positive development. The energy economic index and energy environment index have a relatively gentle multi-peak characteristic and a thicker tail than the normal distribution, namely, the distribution of “wide peak and thick tail”, indicating the emergence of multipolar differentiation. The global energy security index and energy supply index generally follow a unimodal distribution, meaning that there is no obvious multilevel differentiation or heterogeneous groups. However, there is a large unimodal width, indicating that the indices vary greatly. More years fall into the middle and low energy security indices. Meanwhile, the peak value of the energy supply index curve is relatively high, indicating that the global energy supply is gradually improving.

### 4.2. Three-Dimensional Kernel Density Estimates for Global Energy Security

To a certain extent, regional energy security depends on the stability of the energy supply system, the balance of energy service facilities, the sustainability of energy consumption patterns and the superiority of the external environment for energy industry development. From the spatial perspective, its distribution characteristics are basically consistent with the international geopolitical pattern. To further explore the driving mechanism of the optimization of the world energy security pattern since 1995, this section analyzes the density distribution curve of the composite energy security index and other four dimensions from 1995 to 2019 (Figure 15).

(1) The kernel density estimates of the global energy security supply index are shown in Figure 15a. On the whole, the kernel density curves of the energy supply index are characterized by unimodal distribution, suggesting no obvious gradations of global energy supply. Meanwhile, due to the development of the world energy market and the economization and intensification of the regional energy systems, the average value of energy supply security index of global countries is basically stable at about 0.07, showing no large fluctuations in the energy supply security level. From the spatial curve extension, the energy security supply index shows a rising peak height and a decreasing peak width and the right tail of gradually shortens and moves to the left. From the change in the density distribution curve, the range of energy security supply index of increasingly more countries is between 0.04 and 0.1 and the number of countries in this range has increased from 39 in 1995 to 56 in 2019, mainly European developed or high-income countries. This also reflects that developed or high-income countries occupy a favorable position in the global energy market by virtue of their advantages in the market, system, political environment and discourse. In the process of economic globalization, the penetration of external energy markets has effectively met local energy demand, thus guaranteeing the security and stability of the regional energy supply. For example, two oil crises in the 1970s prompted advanced European economies to shift to an energy-saving and low-carbon development model and to guarantee the steady growth of energy supply through strategic energy reserves and enhanced energy diplomacy.

(2) The kernel density estimates of the global energy security use index are shown in Figure 15b. On the whole, the kernel density curves of the energy use index change gradually from multimodal distribution to unimodal distribution, indicating that the bifurcation of the energy security use index is gradually disappearing. Regarding the spatial curve extension, the peak change trend of energy security use index is the same as that of the supply index. From the change of density distribution curve, the number of countries with energy use security index greater than 0.028 has also increased from 53 in 1995 to 75 in 2019. Notably, economically developed countries predominate among these countries. For example, there are 35 EU or OECD countries among the 75 countries with an energy use security index greater than 0.028 in 2019. The cause lies in the strong environmental awareness of the countries and they have a long-term focus on energy efficiency, renewable energy, bioeconomy and natural carbon sinks, carbon capture and storage, leading to a level of sustainability of energy use significantly ahead of the rest of the world. Relying on abundant hydro resources and the promotion of hydropower development technologies, the level of energy use security in Latin America has also increased significantly, with 13 countries having an energy use security index of 0.028 or higher.

(3) The kernel density estimates of the global energy security economic development index are shown in Figure 15c. On the whole, the kernel density curves of this index are characterized by bimodal distribution, while this distribution characteristic is gradually disappearing. In the last five years of the 20th century, the global Energy Security Economic Development Index showed an obvious bimodal distribution, but many countries undertook industrial upgrading and technological innovation. By 2019, its polarization has been effectively improved. For example, the number of countries with an energy economic development index greater than 0.025 has also increased from 53 to 68, mainly located in the Middle East and Eastern European countries with relatively abundant hydrocarbon resources, as well as Latin American countries with abundant hydropower resources. Due to the cheaper cost of energy development technology, as well as improved energy infrastructure and relatively low energy prices, these countries have seen a continuous improvement in their energy economy and technology levels. From the spatial curve extension of the kernel density curves, the peak height decreases and the peak width grows wider, reflecting the widening gap in economic development between countries, and the overall distribution tends to be dispersed. In the past 25 years, most countries with high energy security economic development index have concentrated in North America, Europe, the Middle East and other developed countries. Owing to the higher levels of industrialization and urbanization, the energy consumption demand in these countries is high. However, many countries have actively carried out industrial upgrading, structural upgrading and scientific and technological innovation, thereby reducing the intensity of energy consumption and relieving the pressure on energy supply and demand. From the change in the density curve, the average value of the global energy economic development index increased from 0.034 to 0.044 during the sample period. Attributing to the gradual easing supply and demand, relatively stable or even decreasing average global energy prices and growing levels of national economies and residents’ income, the level of energy economy and technology in developed and emerging economies, which are highly dependent on imported energy, has been improved. For example, the economic development of countries such as South Korea, China and Thailand has been significantly improved.

(4) The kernel density estimates of the global energy security environmental index are shown in Figure 15d. Morphologically, the kernel density curves of this index have obvious double peaks from 1995 to 2019, indicating a serious polarization of the environmental index among countries without effect improvement. It can be seen that the energy consumption of developing countries is lower than that of developed countries due to the lower levels of industrialization and economic development, thus relatively reducing the emission of environmental pollutants; on the other hand, it also reflects the high dependence on global countries on the use of fossil fuels, which is not yet matched by sufficient production of renewable energy. From the spatial curve extension, the kernel density curve shows a gradually rising peak height, which indicates that, although the gap in environmental indices between countries is narrowing, the polarization still exists. From the change of the density curve, the sustainability and efficiency of energy use receive the most international attention. The global energy security environmental index shows a decreasing trend, with its mean value decreasing from 0.04 in 1995 to 0.037 in 2019. Specifically, the number of countries with energy security environmental index greater than 0.04 has decreased from 45 to 38.

## 5. Conclusions and Discussion

### 5.1. Conclusions

In response to the challenges of climate change, environmental degradation and energy poverty, the goal of all global countries is to build a new energy system that is clean, low-carbon and efficient and to accelerate energy transformation and enhance regional energy security. This paper systematically evaluates the state of energy security for 102 countries from 1995 to 2019, reveals the evolution of the spatial pattern of world energy security and main formation mechanisms and puts forward the focus of future energy geopolitical games. The results are as follows.

(1) On the whole, global energy security shows local turmoil, but the overall energy security index has a “J” shaped trend. This indicates that global energy security can be influenced by geopolitics Since the beginning of the 21st century, the global energy security pattern as a whole has been characterized by optimization. Specifically, the number of “Best” energy security countries presents an obvious upward trend, whereas that of countries with “Worse” energy security has decreased significantly. Among them, “Best” countries have increased from 4 in 1995 to 9 in 2019. However, at different stages of development, global energy security has shown obvious differences; especially since 2010, the optimizing trend of the global energy security pattern is more significant.

(2) The level of comprehensive regional development significantly affects the operational efficiency of their energy systems, so the global energy security pattern and international geopolitical order zoning basically coincide. From the perspective of energy security state, “Best” countries are concentrated in Western Europe and North America economically developed regions, “Better” type countries are mainly distributed in Central Europe, Latin America and Asian high-income regions. Meanwhile, countries with “Good” performance are mainly located in the Middle East, Southeast Europe, where the energy system is in desperate need of transformation and “Worse” and “Poor” countries are concentrated in economically underdeveloped Africa and South Asia.

(3) Over the past 25 years, driven by climate-friendly green technologies and sustained low international oil prices, the developed economies and Latin American countries have significantly increased their energy security level. However, there is little room for developing countries to improve their energy security, due to lower innovation capacity, backward productivity and less disposable income of residents.

(4) The kernel density estimates of the global total energy security index and its dimensional indices do not obey traditional empirical distribution. During the sample period, both the global energy security index and the energy supply index were characterized by unimodal distribution; this indicates that the energy security levels of all countries in these two dimensions are not multipolar, but only more dispersed in distribution. The gradual evolution of the energy security use index from bimodal to unimodal indicates that the energy security levels of all countries in this dimension have been effectively improved. However, there are obvious bimodal characteristics of the energy economic development index and energy environment index, indicating that the global economic level is seriously differentiated and the ecological environment has not been improved. Meanwhile, against the background of the rising populism in some countries, the intensification of trade barriers and the strategic contraction of the United States in the Middle East, the future global energy security pattern will be affected by the impact of China–US relations, the pressure on energy transition and the uncertainty of the political environment.

### 5.2. Discussion

Based on the above conclusions, we summarize the challenges that the future global energy security landscape may face as follows:

(1) The global energy supply and demand pattern is gradually changing. In the past decade, due to the rapid rise in oil and gas production brought about by the “shale revolution”, the United States is seeking to regain its place as the world’s energy hegemon. Meanwhile, the rise of emerging economies in East Asia has shifted the center of gravity of world energy consumption eastward and China has now become the world’s top energy demand country. In this context, China and the United States constitute the largest corresponding relationship between supply and demand and the competition and cooperation between two countries have also become the focus of the change of supply and demand in the global energy market.

(2) The growth rate of the global economy in 2019 was 2.3%, lower than that in 2018, suggesting a slowing trend. However, the total energy consumption increased by 1.3% and consequently energy intensity recorded the smallest decline in the past 10 years. In addition, the repeated global outbreaks of energy populism and power competition are challenging the existing rules of multilateral consultation and international order and both the global energy transition and international cooperation are facing more risks or uncertainties.

(3) The risk to social stability caused by energy transition gradually becomes one of the important factors affecting regional energy security in terms of energy economy and technology. Accelerating energy decarbonization is inevitable for all countries in order to effectively respond to climate change and achieve the UN sustainable development goals. However, from the experience of developed countries including Europe and the United States, in the social context of weak economic growth, widening gap between rich and poor and high debt ratio, the increased economic burden of residential energy consumption greatly affects the effectiveness of regional energy transition.

## Figures and Tables

**Figure 1 ijerph-19-14584-f001:**
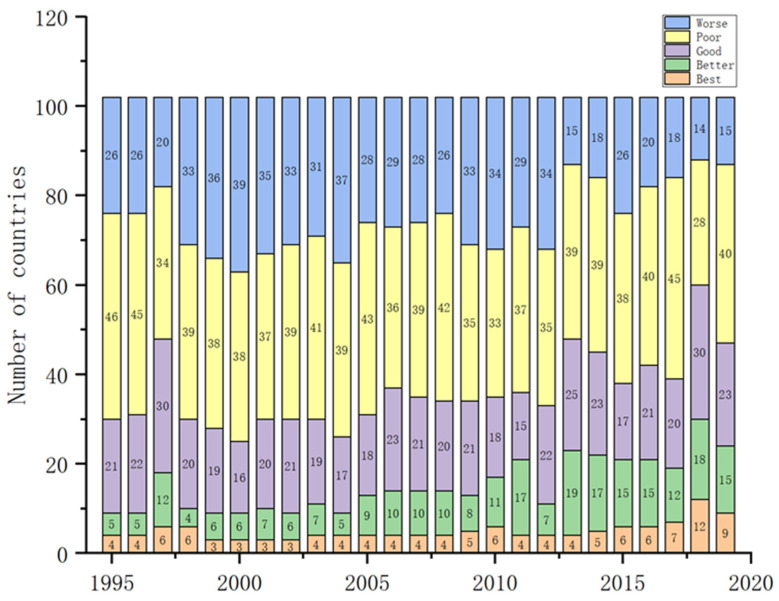
Distribution of energy security status of 102 countries from 1995 to 2019.

**Figure 2 ijerph-19-14584-f002:**
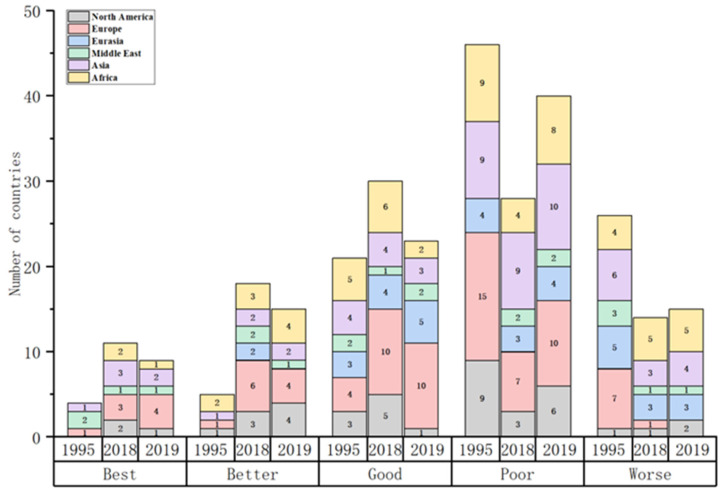
The dynamic evolution of global energy security.

**Figure 3 ijerph-19-14584-f003:**
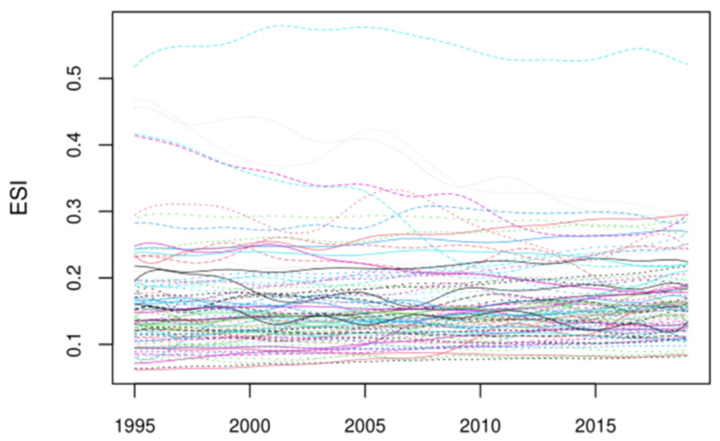
Function curve of global energy security.

**Figure 4 ijerph-19-14584-f004:**
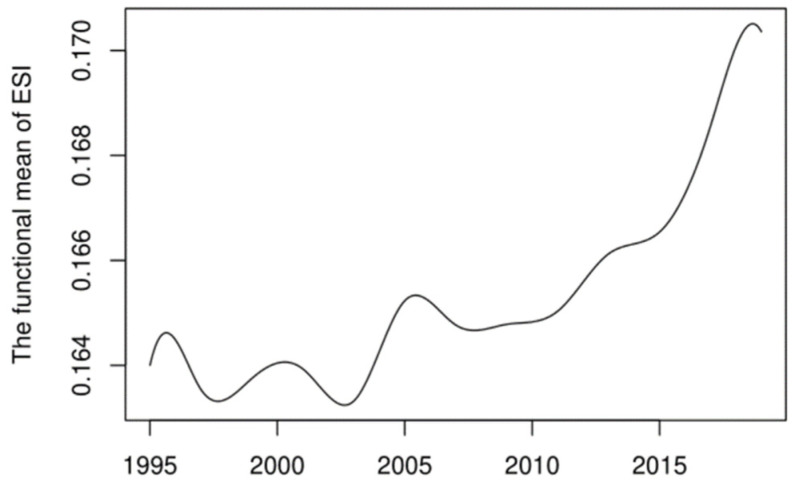
The function curve of energy security means value.

**Figure 5 ijerph-19-14584-f005:**
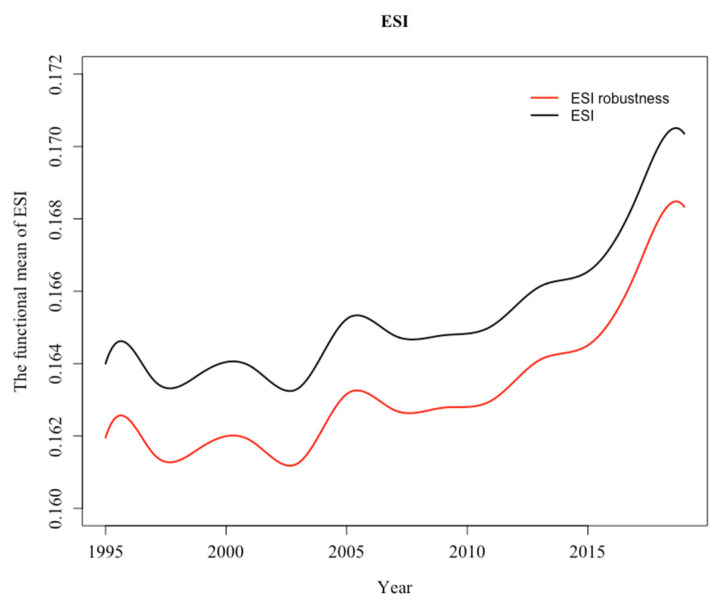
ESI curves for Global corresponding to two methods.

**Figure 6 ijerph-19-14584-f006:**
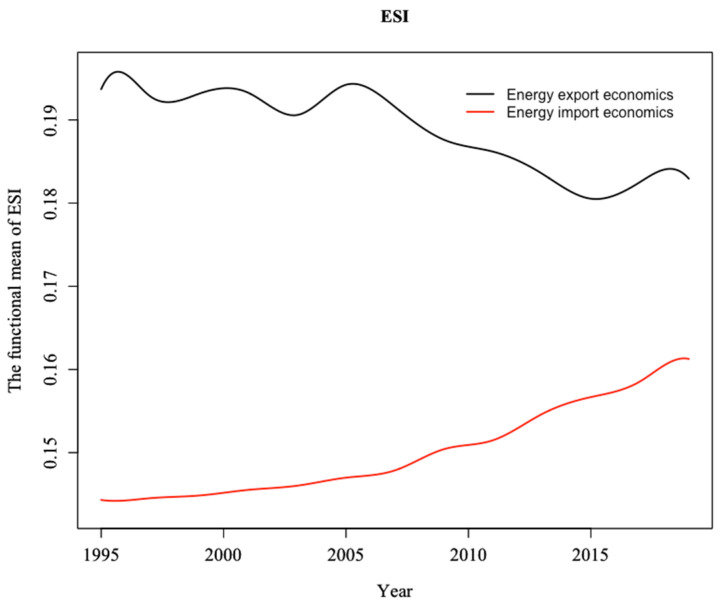
Trends in energy security indices for energy importing and exporting economics.

**Figure 7 ijerph-19-14584-f007:**
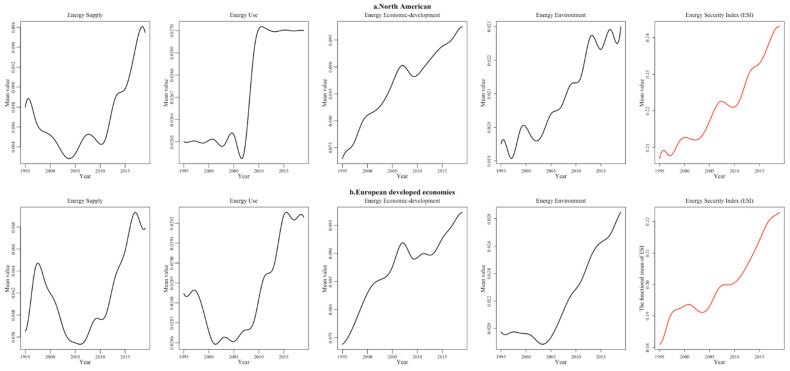
“Best” regional energy security system performance in four dimensions.

**Figure 8 ijerph-19-14584-f008:**
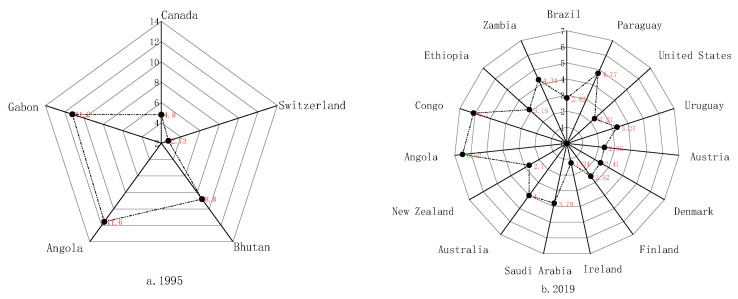
Energy supply index to use index ratio.

**Figure 9 ijerph-19-14584-f009:**
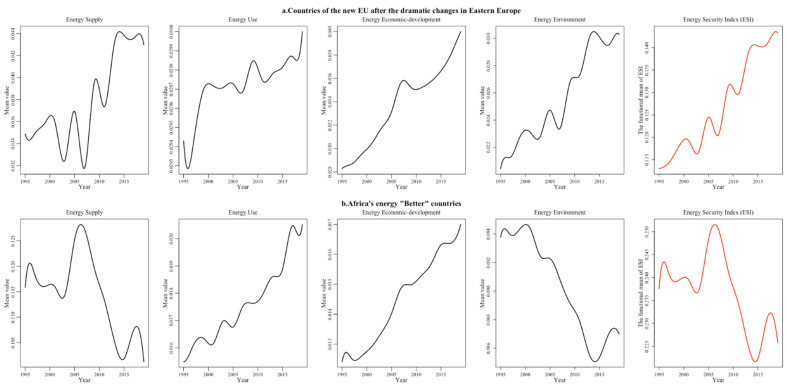
“Better” regional energy security system performance in four dimensions (part 1).

**Figure 10 ijerph-19-14584-f010:**
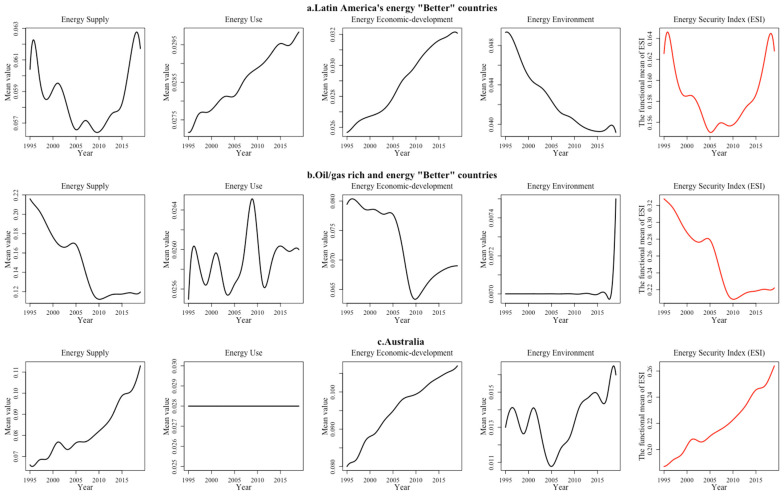
“Better” regional energy security system performance in four dimensions (part 2).

**Figure 11 ijerph-19-14584-f011:**
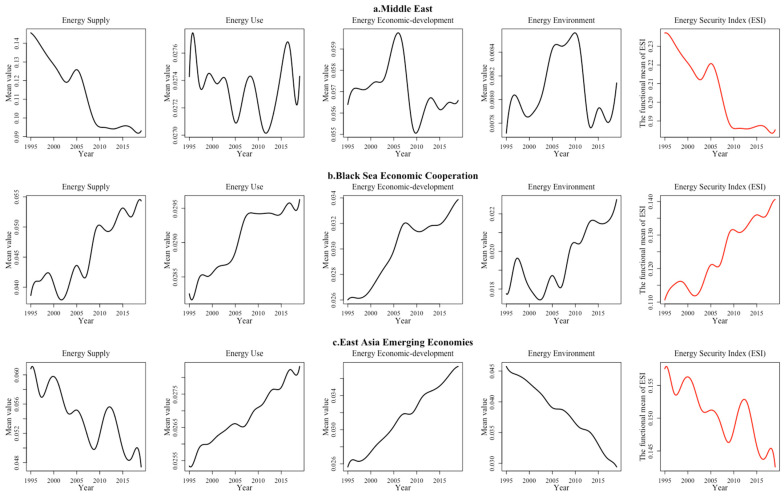
“Good” regional energy security system performance in four dimensions.

**Figure 12 ijerph-19-14584-f012:**
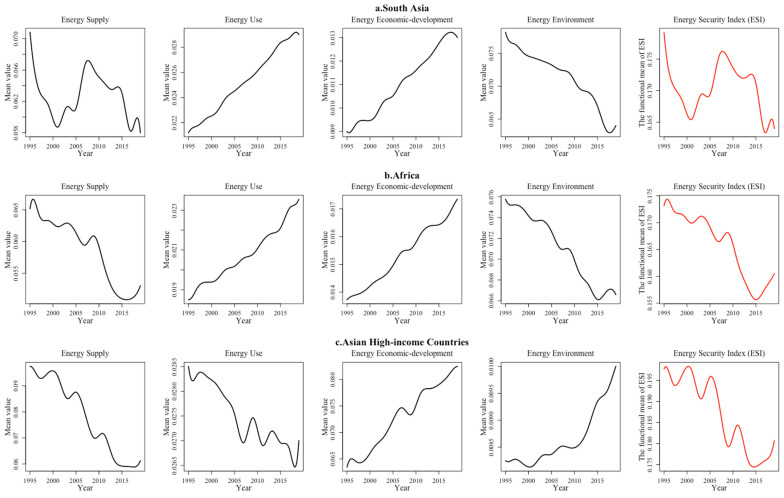
“Poor” and “Worse” regional energy security system performance in five dimensions.

**Figure 13 ijerph-19-14584-f013:**
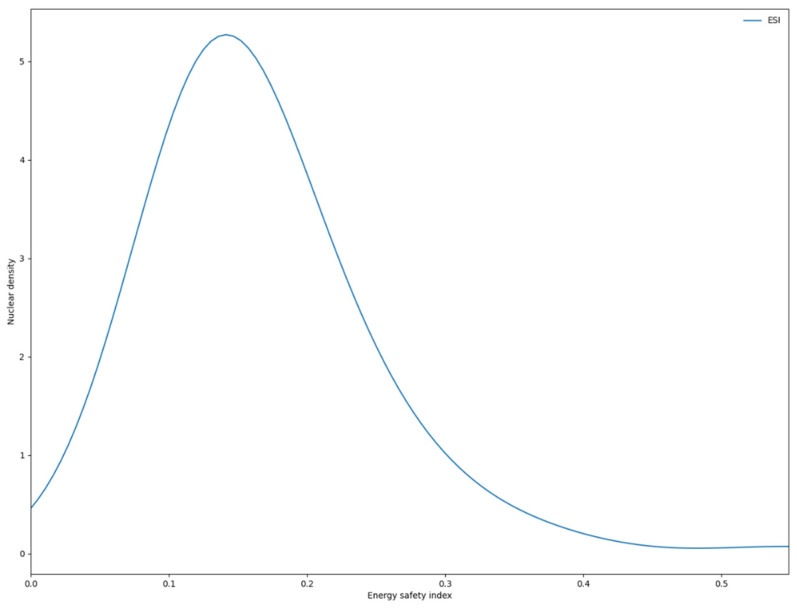
Kernel density estimates for the global energy security total index.

**Figure 14 ijerph-19-14584-f014:**
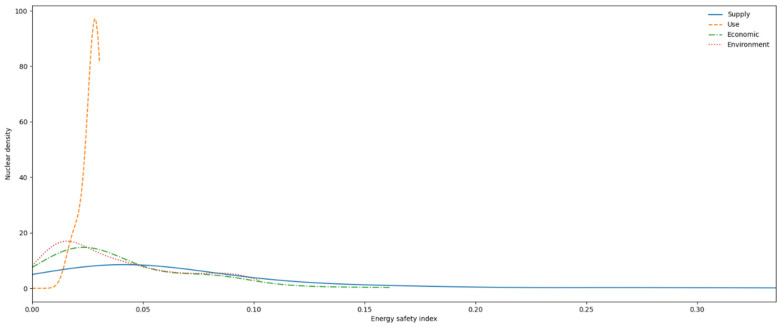
Kernel density distribution in different dimensions of global energy security.

**Figure 15 ijerph-19-14584-f015:**
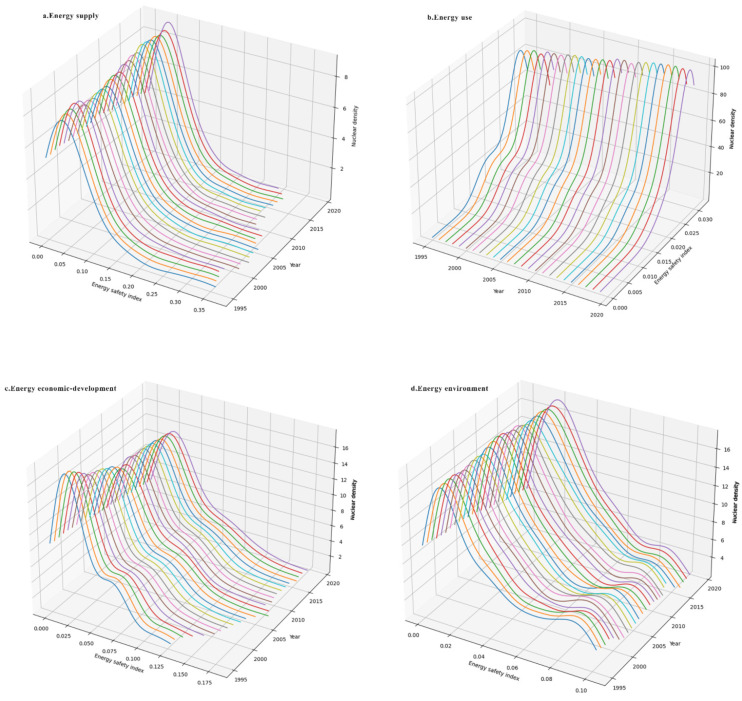
Evolution of energy security index in different dimensions.

**Table 1 ijerph-19-14584-t001:** Selected indicators for the dynamic evolution of global energy security in different dimensions.

Dimension	Code	Indicator	Description (Unit)	Symbol
Supply security	ES1	Energy self-sufficiency rate	Energy production/Total energy consumption (%)	+
	ES2	Share of electricity generation	Electricity generation/Total energy consumption (%)	−
	ES3	Share of renewable energy generation	Renewable energy generation/Total electricity generation (%)	+
	ES4	Share of non-fossil fuel	1-Fossile fuel/Total energy consumption (%)	+
	ES5	Share of Fuel imports	Fuel imports/Total energy consumption (%)	−
	ES6	Energy production per capita	Energy production/Total population (MMBtu/person)	+
Using security	EU1	Electricity intensity	Electricity generation/GDP (kWh/GDP)	−
	EU2	Energy consumption per capita	Energy consumption/Total population (MMBtu/person)	−
	EU3	Energy intensity	Energy consumption/GDP (1000 Btu/GDP)	−
	EU4	Electricity per capita	Electricity generation/Total population(kWh/person)	−
	EU5	Energy consumption elasticity Factor	Energy consumption growth rate/GDP growth rate	−
	EU6	Energy Population elasticity Factor	Energy consumption growth rate/Population growth rate	−
	EU7	Share of energy depletion	Energy depletion/GIN (%)	−
	EU8	Access to electricity	The percentage of the population with access to electricity	+
Economic-development security	ED1	Industrialization level	Industrial value-added/GDP (%)	+
	ED2	Urbanization level	Urban population/Total population (%)	+
	ED3	Population growth rate	A given year’s Population/Previous year’s population-1	−
	ED4	GDP per capita	GDP/Total population (USD)	+
	ED5	Electricity production elasticity coefficient	Electricity production/GDP growth rate	−
	ED6	GDP growth rate	(A given year’s GDP -Previous year’s GDP)/Previous year’s GDP	+
Environmental security	EE1	NO emissions per capita	NO emissions/Total population (Kg/person)	−
	EE2	Carbon Intensity	CO_2_ emissions/GDP(Mt/GDP)	−
	EE3	Share of particulate emissions damage	Particulate emissions damage/GIN (%)	−
	EE4	Share of clean energy	Renewable energy production/Total energy consumption (%)	+

**Table 2 ijerph-19-14584-t002:** Changes in the number of different types of energy security globally.

Type	1995	2000	2010	2018	2019	Changes from 1995~2000	Changes from 2010~2018	Changes from 2019
Best	4	3	6	11	9	−1	3	−2
Better	5	6	11	18	15	1	7	−3
Good	21	16	18	30	23	−5	12	−7
Poor	46	38	33	28	40	−8	−5	12
Worse	26	39	34	14	15	13	−20	1

## Data Availability

The datasets used in this study are available from the corresponding author on reasonable request.

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
