# Peer review of "The Dynamic Evolution of Global Energy Security and Geopolitical Games: 1995~2019"

_ijerph, 2022, doi:10.3390/ijerph192114584_

Round 1

Reviewer 1 Report

The paper evaluates the global energy security and tries to explain the issue from the aspect of geopolitical games. Based on EWM, it provides an index to measure the energy security situation in different countries. The paper could be relevant from the energy policy-making perspective. Nevertheless, as it stands, it needs thorough and substantial revisions in terms of both form and content, as follows: 

1.       The authors need to better illustrate their paper and underline its novelty compared to previous research in the sector of Introduction. Additionally, the dynamic evolution of global energy security might be too simple, it must be proved by some time-vary relationship or temporal-correlation indeed.

2.       The paper use entropy weighting method (EWM) to analysis the issue of energy security. Although EWM is subjective enough, the authors should change other methods to confirm the robustness of result which shows “J” shape trend.

3.       It is necessary to consider the heterogeneity between energy export economies and energy import economies in the sector of discussion of result.

4.       Last but not least, there are many style and grammar issues left such as “has” in line 9. Thus, we recommend the authors to resort to professional proofreading services.

Reviewer 2 Report

Reasonable effort has been made by the authors and I feel it is valuable to the research community. The manuscript is fine but I feel some of the improvements must be made as followed.

1) Pg 3 line 115, “AR” appeared all of sudden. Please include the full form of AR.

2) Pg 3 line 138, “when establishing… principal component analysis”, references of the studies may be mentioned such as https://doi.org/10.3390/en15113929,

 https://doi.org/10.1016/j.esr.2021.100710, https://doi.org/10.3390/su14106099.

3) Pg 3 line 144, “FDA focuses on curved data …. which was rarely used “, please clarify and explain about what is the main advantage of using the curved data in your case?

4) Pg 5 line 187, line ending “on the curvilinear form”, please add the explanation that the data utilized in your study has actually attained curvilinear form.

5) Section 2.5, Table 1, please insert references for dimensions and indicators sources.

6) Section 5, pg 19, “J-shape trend” is mentioned. Please add explanation about what J-shape is revealing?  

Round 2

Reviewer 1 Report

Agree to publish.